# Diet-Wide Association Study for the Incidence of Type 2 Diabetes in Three Population-Based Cohorts

**DOI:** 10.3390/nu16223798

**Published:** 2024-11-05

**Authors:** Hye Won Woo, Manh Thang Hoang, Min-Ho Shin, Sang Baek Koh, Hyeon Chang Kim, Yu-Mi Kim, Mi Kyung Kim

**Affiliations:** 1Department of Preventive Medicine, College of Medicine, Hanyang University, Seoul 15588, Republic of Korea; woohehe@hanyang.ac.kr (H.W.W.); 2Institute for Health and Society, Hanyang University, Seoul 15588, Republic of Korea; 3Wolfson Institute of Population Health, Queen Mary University of London, London E1 4NS, UK; 4Department of Preventive Medicine, Chonnam National University Medical School, Gwangju 61469, Republic of Korea; 5Department of Preventive Medicine and Institute of Occupational Medicine, Yonsei Wonju College of Medicine, Wonju 26493, Republic of Korea; 6Department of Preventive Medicine, Yonsei University College of Medicine, Seoul 03722, Republic of Korea

**Keywords:** diet-wide association study, beans, fruits, type 2 diabetes, interrelationship, population-based cohort

## Abstract

Background: Dietary factors are well-known modifiable risk factors for type 2 diabetes (T2D), but many studies overlook the interrelationships between these factors, even though foods are often consumed together and contain a variety of nutrients. Objectives: In this study, we employed a diet-wide association study approach to investigate the links between various dietary factors and T2D onset, taking into account complex dietary patterns. Methods: We analyzed 16,666 participants without T2D from three Korean population-based cohorts: the Multi-Rural Communities Cohort (*n* = 8302), the Atherosclerosis Risk of a Rural Area Korean General Population cohort (*n* = 4990), and the Kanghwa cohort (*n* = 3374). A two-step approach was employed. In the first step, robust Poisson regression analysis was used for the initial screening (false discovery rate-adjusted *p*-values < 0.05). In the second step, a hierarchical cluster analysis was conducted of all dietary factors, followed by mutual adjustment of the screened factors within each cluster to account for interrelationships. Results: The 11 food clusters screened were cooked rice with beans, rice cakes, breads/spreads, bread products, cheese and pizza/hamburger, grain powder, snack/confections, nuts and roasted beans, soy milk, traditional beverages, and non-native fruit. These factors were similarly distributed across three of the seven clusters in each cohort. After mutual adjustment, cooked rice with beans (*p*-value ≤ 2.00 × 10^−7^ in all three cohorts) and non-native fruits (*p*-value ≤ 5.91 × 10^−3^ in two cohorts) remained significantly associated with lower T2D risk in more than one cohort. Conclusions: The inverse association of cooked rice with beans, not observed with other types of cooked rice, and that of non-native fruits, suggest that incorporating beans into rice and eating various fruits may be an effective strategy for preventing diabetes.

## 1. Introduction

Type 2 diabetes (T2D) has emerged as a global health crisis, affecting 9.3% of the world’s population [1]. Its rapid increase has made it the seventh leading cause of disability-adjusted life years and the ninth leading cause of mortality, outpacing both population growth and aging [2]. Factors such as the highly processed, calorie-rich Western diet and a sedentary lifestyle are possible explanations for this increase [2]. A healthy diet has become an increasingly crucial basis for prevention of T2D [3].

The critical role of diet in T2D risk is well established [4]. Traditional research approaches have often focused on isolated dietary components. However, this reductionist method fails to capture the intricate interplay between dietary factors, and has led to fragmented and inconsistent findings across studies [5]. To address these limitations, researchers have suggested dietary pattern approaches, such as the Mediterranean diet and the Alternative Healthy −ating Index (AHEI) [6]. However, these methods aggregate foods into broad categories, potentially masking the specific impacts of individual foods and nutrients. Consequently, there is a significant gap in understanding the impacts of individual dietary components and their interactions on T2D risk in real-world eating patterns. Given the complexity of diets and the potential for synergistic and antagonistic effects between foods/nutrients, a more nuanced analytical approach is required to clarify the true association between diet and T2D development.

Recently, a novel approach known as Diet-Wide Association Studies (DWAS) has been adapted from genome-wide association studies [7,8]. DWAS allow simultaneous evaluation of numerous foods and nutrients. However, previous DWAS relied mainly on Spearman correlations between dietary factors and presented results using Manhattan plots based on food groups. However, these analyses did not account for potential interactions between dietary factors, such as the synergistic or antagonistic effects that may occur when nutrients are consumed together. [7,8]. While statistical adjustments like the false discovery rate (FDR) in DWAS can partially address the issue of multiple comparisons, they do not resolve the failure to account for these interrelationships. Currently, DWAS for diabetes have been conducted only in Western populations [7,8]. There is a need for similar studies in Asian populations, because differences in food cultures and health profiles between Western and Asian populations may result in different dietary impacts on diabetes risk.

This study employed a novel two-step approach to systematically analyze the interrelated dietary components associated with T2D risk. First, we conducted a diet-wide association study on 47 predefined foods/food groups and 62 nutrients across three prospective cohorts of adults aged 40 years or older to screen for associations with T2D incidence. Next, we applied hierarchical cluster analysis to group closely related dietary factors, analogous to grouping SNPs into natural bins based on chromosomal locations, to clusters of closely related dietary components. In addition to identifying the lead signal within each cluster, we also obtained marginal associations after adjusting for intra-cluster relationships accounting for the intricate interdependencies among dietary components.

## 2. Materials and Methods

### 2.1. Study Population and Design

The study population comprised adults aged 40 years and older from three population-based cohorts in South Korea. The Multi-Rural Communities Cohort (MRCohort), consisting of participants from three regions—Yangpyeong, Namwon, and Goryeong—enrolled 9759 participants. The Atherosclerosis Risk of a Rural Area Korean General Population (ARIRANG) cohort, spanning two regions—Wonju and Pyeongchang—included 5942 participants, and the Kanghwa cohort, based in a single region (Kangwha), comprised 3845 participants. All participants were free of cardiovascular disease and cancer at baseline, were enrolled between 2005 and 2011, and were followed from 2007 to 2017. The follow-up rates for MRCohort and the ARIRANG and Kanghwa cohorts were 81.0%, 77.6%, and 72.1%, respectively, with participants attending one or more follow-up visits.

For this study, we excluded participants who reported the use of antidiabetic medications or insulin at baseline or whose fasting blood glucose (FBG) level at baseline was ≥126 mg/dL (7.0 mmol/L) (*n* = 2156). Additional exclusions were (1) participants who left more than 10 items blank on the Food Frequency Questionnaire (FFQ), (2) those with implausible total energy intake (≥99.5th percentile or ≤0.5th percentile of total energy intake) (*n* = 284), and (3) those with missing data on covariates (education level, regular exercise, smoking status, and/or alcohol consumption, *n* = 440). After these exclusions, the final analysis included 8302 participants from the MRCohort, 4990 from the ARIRANG cohort, and 3374 from the Kangwha cohort. The study was conducted following the principles of the Declaration of Helsinki, and the study protocol was approved by the following Institutional Review Boards: Hanyang University, Chonnam University, and Keimyung University for the MRCohort; Yonsei Wonju College of Medicine for the ARIRANG cohort; and Yonsei University for the Kanghwa cohort. All participants provided written informed consent before participating in the study.

### 2.2. Dietary Assessments, Food Groups, and Nutrients

Dietary intake was assessed at baseline, and follow-up surveys using the same validated 106-item Food Frequency Questionnaire (FFQ) were administered by trained interviewers. The FFQ consists of food items with nine frequency categories ranging from ‘never or rarely’ to ‘three times/day’ and three portion sizes specified for each item [9]. Daily food consumption (serving/day) was calculated by multiplying the weighted frequency by the average serving size for each food. To determine optimal food groupings, we initially reduced the 106 food items to 104 by aggregating five rice-related items (cooked white rice, cooked rice with beans, cooked rice with multi-grains, cooked white rice only or with beans, and cooked rice with beans or with multi-grains) into three categories (cooked white rice, cooked rice with beans, and cooked rice with multi-grains). We then employed Ward’s hierarchical clustering method in conjunction with the silhouette method, running analyses with predefined numbers of clusters (20, 30, and 40). The interpretability of these clusters was examined to ensure consistency with the 37 predefined food groups used in previously published studies [10], and we ultimately identified 47 modified food groups (thereafter food) that best represented the dietary patterns in our study population (Appendix A). Nutrient intakes were calculated using the nutrient database developed by the Korean Nutrition Society, which is based on the seventh edition of the Korean Food Composition Table [11], and an established antioxidant database [12]. To account for long-term dietary patterns and minimize within-person variation [13], we calculated cumulative average consumption by averaging daily food and nutrient consumption across all visits up to the endpoint or censoring. All food and nutrient intakes were adjusted for total energy intakes using the residual method [14].

### 2.3. Assessment of Nondietary Factors and Health Examination

Trained interviewers collected comprehensive data on nondietary factors using standardized protocols and structured questionnaires. The information gathered included demographics (age, sex, and area of residence), socioeconomic status (educational level), lifestyle factors (physical activity, smoking status, and alcohol consumption), and medication use (antihyperglycemic and insulin). Higher education was defined as completion of 12 or more years of schooling, while regular exercise was more than three 30 min sessions per week. Daily alcohol consumption (g/day) was calculated by multiplying the average frequency of consumption of any of six types of alcoholic beverage (soju, takju, beer, refined rice wine, wine, and whisky; times/day) by the amount of each beverage consumed per occasion, factoring in the alcohol content (%) of each beverage and using alcohol gravity (0.7947).

At every health examination, anthropometric data were measured. Height was measured using a standard height scale to the nearest 0.1 cm, and weight was measured using a metric weight scale to the nearest 0.01 kg while wearing light clothes without shoes. Body mass index (BMI) was calculated by dividing weight in kilograms by height in meters squared. Biochemical data, such as fasting blood glucose, were analyzed in serum collected after at least eight hours of fasting using an ADVIA 1650 Automated Analyzer (Siemens, New York, NY, USA).

### 2.4. Ascertainment of Diabetes Incidence

At each visit, the participants reported any physician-diagnosed T2D and use of antidiabetic medications or insulin. For those without T2D at baseline, we defined new cases as either participants newly diagnosed by a physician who initiated antidiabetic medication or insulin treatment or those with fasting blood glucose ≥126 mg/dL (7.0 mmol/L) during follow-up examinations, in accordance with the American Diabetes Association guidelines [15]. We calculated follow-up time for each participant from their enrollment date until T2D diagnosis or study end in December 2017, whichever occurred first. For the participants lost to follow-up, we assigned half the median follow-up time of those who were successfully followed up [16].

### 2.5. Statistical Analysis

The general characteristics across the three cohorts were summarized using means for the continuous variables and percentages for the categorical variables. To identify dietary factors (among 47 foods and 62 nutrients) associated with T2D incidence, we employed a two-step approach. In the first step, we conducted an initial screening step by calculating the incidence rate ratios (IRRs) for each food and nutrient using modified Poisson regression with robust error estimation [17,18]. Covariates, including age (years), sex, higher education level (≥12 years), regular exercise (≥3 times/week for ≥30 min/session), current smoking status, alcohol consumption (g/day), and total energy intake (kcal/day), were adjusted in these models. The participants were grouped into tertiles of food and nutrient intake, with the IRRs presented using the lowest tertile as the reference group. Dietary factors were designated as ‘validated dietary factors’ if they achieved statistical significance in the highest tertile with FDR-adjusted *p*-values < 0.05 in more than one cohort. In the second step, we examined the interrelationships among these validated dietary factors. Partial Spearman correlation coefficients were computed for all standardized dietary factors, adjusting for the same set of covariates. These dietary factors were visualized using hierarchical clustering, which resulted in seven distinct clusters. The associations between foods and nutrients and T2D risk found in the first screening step were presented in a Manhattan plot according to clusters. In the second step, we ran additional models that mutually adjusted for validated dietary factors within each cluster to account for potential correlations among dietary factors within the same cluster. This allowed us to further investigate the marginal associations of individual dietary factors.

Statistical analyses were conducted using two software packages. The cluster analysis and Manhattan plots construction were performed using R statistical software (version 4.1.0; R Foundation for Statistical Computing, Vienna, Austria). All other statistical analyses were executed using SAS software (version 9.4; SAS Institute Inc., Cary, NC, USA).

## 3. Results

Table 1 presents the general characteristics of the three cohorts. Notable differences were observed in mean age (MRCohort being the oldest at 61.3 years), education level (ARIRANG having the highest percentage of individuals with higher education at 35.9%), and the lifestyle factor of regular exercise (highest in Kanghwa at 24.3%), while BMI and total energy intake were similar across the cohorts. Regarding dietary factors, the MRCohort had the highest intake of cooked white rice, Kanghwa reported the highest intake of cooked rice with beans, and fruit consumption was highest in Kanghwa, while ARIRANG had the lowest intake of fruit (Appendix A).

During a total follow-up period of 96,609 person-years across the three cohorts, we observed 536 T2D events in the MRCohort (median follow-up: 6.2 years; interquartile range [IQR]: 3.5–8.9), 253 events in the ARIRANG cohort (median follow-up: 6.3 years; IQR: 3.1–8.8), and 164 events in the Kanghwa cohort (median follow-up: 4.9 years; IQR: 1.2–7.6). The initial screening revealed significant associations (FDR-adjusted *p*-values < 0.05) for 18 foods and four nutrients in the MRCohort, 11 foods and two nutrients in the ARIRANG cohort, and nine foods in the Kanghwa cohort. The directions of these associations were consistent across the three cohorts. Among these significant dietary factors, only 11 foods (not nutrients) were validated in at least two cohorts (>1 cohort at original *p*-value = 7.15 × 10^−3^): cooked rice with beans (RiceBeans), rice cakes (RiceCakes), breads/spreads (BreadSpread), bread products (BreadProduct), cheese and pizza/hamburger (ChzPizzaBurger), grain powder (GrainPowder), snack/confections (SnackConfect), nuts and roasted beans (NutsRoastBeans), soy milk (SoyMilk), traditional beverages (TradBeverages), and non-native fruit (NonNatFruit), referring to fruits not traditionally cultivated in Korea but imported for consumption (Table 2).

In the hierarchical cluster analysis to identify interrelated dietary factors, seven distinct food/nutrient clusters, including dietary factors that were not identical but very similar, were extracted in all three cohorts (Appendix A). We grouped the seven clusters of similar dietary factors as follows (see Appendix A): Group 1_Carbohydrate nutrition factors (includes carbohydrates, glycemic load, glycemic index, cooked white rice (WhiteRice), etc.); Group 2_Animal-based nutrients (includes protein, vegetable protein, vitamin D, calcium, milk/yogurt (MilkYogurt), etc.); Group 3_Diverse plant-based foods and seafood (includes NutsRoastBeans, NonNatFruit, GrainPowder, SoyMilk, etc.); Group 4_Rice with beans and related nutrients; Group 5_Plant-based nutrients and vegetables (includes kimchi, fiber, vitamin B6, folate, etc.); Group 6_Green tea, related phytochemicals (includes green tea, copper (Cu), flavan-3-ols), and n-3 polyunsaturated fatty acids; and Group 7_Fusion diet combining traditional and non-traditional foods (includes SnackConfect, RiceCakes, TradBeverages, ChzPizzaBurger, BreadSpread, etc.). In the ARIRANG cohort, Clusters 3 and 2 were classified as the same group (Group 3) in light of the patterns of the other cohorts, but they were further sub-divided into Groups 3a and 3b. The associations of 109 dietary factors and T2D risk are shown in a Manhattan plot according to group in Figure 1.

Table 3 presents the associations between the consumption of the 11 validated foods and the incidence of T2D, with the data mutually adjusted for interdependencies within food clusters across the three cohorts. In the first step, the 11 foods were classified into only three groups (Group 7, Group 3, and Group 4). Cooked rice with beans (RiceBeans) in two cohorts and non-native fruit (NonNatFruit) and grain powder (GrainPowder) in one cohort were not classified with any other validated foods. Among the groups that included two or more validated factors, traditional beverages (TradBeverages) in the MRCohort, cooked rice with beans (RiceBeans) in the ARIRANG cohort, and rice cakes (RiceCakes) in the Kanghwa cohort were the most significant within their respective clusters, even after mutual adjustment of the same cluster.

However, in the MRCohort, non-native fruit was the most significant food item following mutual adjustment, replacing soy milk. Finally, applying the same significance threshold and validation criteria as in the screening step, only two foods—cooked rice with beans (RiceBeans, with soybeans being the primary bean used) and non-native fruit (NonNatFruit)—remained significant in at least two of the three cohorts. The IRRs for cooked rice with beans were 0.37 (95% CI: 0.31–0.46; mutually adjusted *p*-value = 5.53 × 10^−22^) in the MRCohort, 0.41 (95% CI: 0.31–0.50; mutually adjusted *p*-value = 1.51 × 10^−8^) in the ARIRANG cohort, and 0.37 (95% CI: 0.25–0.54; mutually adjusted *p*-value = 2.00 × 10^−7^) in the Kanghwa cohort. Those for non-native fruit were 0.73 (95% CI: 0.58–0.91; mutually adjusted *p*-value = 5.91 × 10^−3^) in the MRCohort and 0.62 (095% CI: 44–0.86; mutually adjusted *p*-value = 4.21 × 10^−3^) in the ARIRANG cohort. Although traditional beverages, bread products, and snacks within Group 7 in the MRCohort, and rice cakes in the Kanghwa cohort remained significant within their respective cohorts, they were not considered final validated factors. We conducted stratification analyses of several risk factors to determine whether the associations with cooked rice with beans and non-native fruits were confounded by or interacted with other factors influencing T2D risk; however, the results remained robust (Appendix A).

## 4. Discussion

In this diet-wide association study on T2D conducted across three population-based cohorts analyzing 47 foods and 62 nutrients, we identified 11 foods inversely associated with T2D risk: cooked rice with beans, rice cakes, breads/spreads, bread products, cheese and pizza/hamburgers, grain powder, snacks/confections, nuts and roasted beans, soy milk, traditional beverages, and non-native fruit. These 11 validated foods were grouped into three of seven clusters, which, while not identical, were highly similar. After mutual adjustment within their respective clusters, only cooked rice with beans (RiceBeans) and non-native fruits (NonNatFruit) remained validated. Despite the geographic differences among the three cohorts, we observed consistent, though not identical, dietary patterns (seven clusters), which were categorized into seven groups. These patterns could be further classified into three main categories: (1) staple food-related patterns, including carbohydrate nutrition factors (Group 1), rice with beans and related nutrients (Group 4), and fusion diets combining traditional foods and non-traditional foods (that can substitute for rice, Group 7); (2) side dish-related patterns, including animal-based and plant-based nutrients and foods (Group 2, 3, and 5); and (3) green tea, related phytochemicals, and n3-PUFA (Group 6). In a previous study [10], we identified three dietary patterns using 37 predefined foods based on similarities in nutrient composition and culinary use. The ‘Meat/Poultry/Seafood’ pattern was split into Group 3 (similar to ‘vegetable/seaweeds’) and Group 7 (similar to ‘non-traditional/non-staple foods’) in the present study. This difference is likely due to the greater number of dietary patterns (seven vs. three) and the different statistical methods used (hierarchical cluster analysis vs. factor analysis).

The two studies just referred to had different objectives. The earlier study [10] aimed to extract dietary patterns using factor analysis to assess overall dietary scores, whereas the present study used cluster analysis to identify groups of closely-correlated foods. A key strength of the present study is its data-driven approach to determining the 47 food groups, which minimizes the subjective bias often introduced when experts define food categories based on personal knowledge—a common practice in previous studies [19,20]. However, it remains important to confirm whether the seven patterns identified in the present study are consistent across the general population and in other cohorts.

The inverse association between incorporating beans (including pulses, subgroups of legumes, etc.) into the diet or increasing the ratio of beans to white rice and the risk of T2D is consistent with findings from observational studies [21,22,23,24,25]. This protective association was also identified in a systematic review and meta-analysis of both short- and long-term randomized controlled trials [26]. The beneficial effects of beans may be attributed to their bioactive components, such as plant protein, fatty acids, soy proteins, and isoflavones. These compounds help improve glycemic control, slow carbohydrate absorption, enhance insulin sensitivity [24,25], and accord with the nutrients grouped with cooked rice with beans in the present study, although they did not pass the threshold of the initial screening step. Additionally, cooked rice with beans was negatively correlated with other rice-based dietary factors in the ARIRANG cohort (Appendix A). The combination of rice and beans, a staple in many Asian diets, may offer unique nutritional benefits for preventing T2D [27,28]. This is remarkable, as white rice alone, with its high glycemic index, has been linked to a higher risk of T2D [29,30]. Our results highlight the potential benefits of beans, particularly soybeans, in this context. However, different types of legumes may have varying impacts on T2D risk [23,24]. Therefore, further research is needed to investigate the associations between specific types of legumes and T2D risk to better understand their individual effects and potential mechanisms.

Our study indicated that while native fruit primarily grown and consumed in our population was not associated with T2D risk in any of the three cohorts (Figure 1), a protective association was observed with non-native fruits, including bananas and oranges, and T2D. However, a recent meta-analysis of fruit consumption did not report a significant link between banana, orange, or total fruit intake and risk of T2D [4,31]. Considering that dietary variety has protective effects against metabolic diseases [32], it is plausible that the observed benefit of non-native fruits may reflect a tendency for individuals with higher consumption of non-native fruits to select a wider variety of fruits, potentially leading to overall health improvements.

Previous DWAS on T2D have provided valuable insights, but with limitations. For example, an environment-wide association study [8] found a positive association between vitamin C-tocopherol and T2D prevalence, and a negative association with beta-carotene, in the US population. However, the cross-sectional design of the study limited its ability to yield temporal relationships. Another DWAS, using UK Biobank data, analyzed 224 dietary factors (food and alcohol) and 21 nutrient factors and identified nine dietary factors and one nutrient (iron) associated with T2D risk, but the authors did not perform an integrated analysis of the dietary factors and nutrients [7]. Notably, sliced buttered bread consumption increased risk, and iron, typically associated with increased T2D risk [33], was also positively associated. However, the present study demonstrated a different direction of association. Foods linked to T2D risk between populations may be attributed to their unique culinary traditions and eating patterns. Therefore, directly comparing our findings with those of other studies is challenging. Nevertheless, these discrepancies in the associations of specific dietary factors across populations may suggest the need to interpret and understand these associations within the context of dietary culture.

Previous DWAS on diseases [34,35] other than diabetes have not included mutual adjustment due to its complexity, the traditional focus on individual dietary components, and the difficulty of efficiently managing such complex analyses. Our study advances the field, as it is the first to apply mutual adjustment for related foods and nutrients within the same cluster. This DWAS approach may help determine whether established factors maintain their associations with health outcomes after mutual adjustment, potentially challenging or confirming existing nutritional knowledge. However, given the complexity of diets in which nutrients are consumed in different combinations, and the limitations of self-reported dietary questionnaires, it is difficult to reliably identify the health effects of individual dietary components in observational studies [36]. The DWAS approach offers a powerful tool for nutritional epidemiology, but it should be used to complement rather than replace candidate approaches. Therefore, all initial 11 foods identified (and even nine other foods that did not pass the final threshold of mutual adjustment) may be promising candidates for further studies in each cohort. Therefore, a combined strategy could provide a more comprehensive understanding of diet–disease relationships, balancing the strengths of hypothesis-driven and data-driven research methods [37].

Our study has several limitations. First, as it was conducted using three rural community cohorts, caution should be exercised when generalizing our results to other populations, particularly urban areas, where lifestyle variables associated with T2D might differ [38]. Second, while we considered relationships among foods/nutrients, there could be other non-adjusted confounders or unmeasured factors related to dietary consumption. Moreover, we cannot fully explain the roles of the overlaps common in mutual adjustments between various foods/nutrients in the development of T2D risk. Third, our analytical approach has its own limitations in that we used only findings in the highest tertiles and evaluated only linear associations. Given that the clusters were created using Spearman correlation coefficients, other dimension reduction methods for dietary factors should be applied to identify possible nonlinear associations. These limitations point to the need for further research to validate and expand upon our findings in diverse populations and using more comprehensive analytical approaches.

## 5. Conclusions

In conclusion, this diet-wide association study across three Korean cohorts, analyzing 47 foods and 62 nutrients, identified 11 foods inversely associated with T2D risk. Notably, it appears that incorporating beans into rice and eating various fruits, including non-native fruits, may have a beneficial role in reducing T2D risk. These findings provide valuable insights into the complex interrelation between dietary components and T2D risk, and further studies to confirm our findings are needed to inform future nutritional recommendations for T2D prevention.

## Figures and Tables

**Figure 1 nutrients-16-03798-f001:**
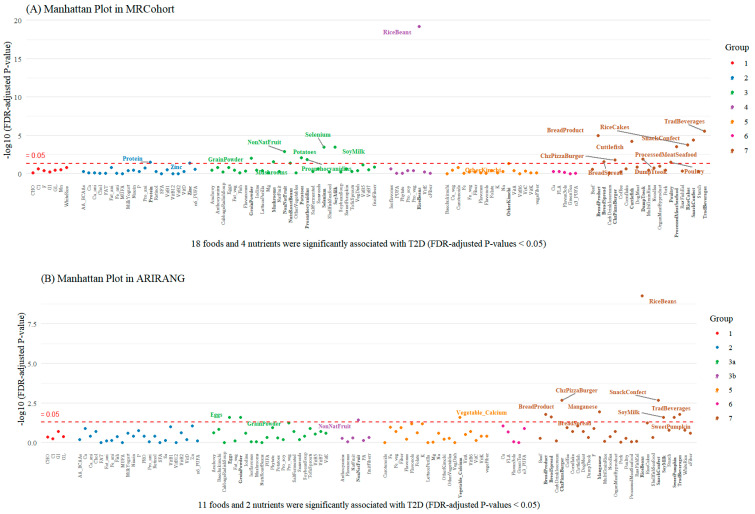
Manhattan plot of associations of foods and nutrients with type 2 diabetes in the three cohorts: (**A**) MR Cohort, (**B**) ARIRANG, and (**C**) Kanghwa cohort. **Abbreviations:** WhiteRice, cooked white rice; RiceBeans, cooked rice with beans; MultiGrainRice, cooked rice with multi-grains; Noodles, noodles; DumpTteok, dumpling and teokguk; RiceCakes, rice cakes; Cornflake, cornflakes; BreadSpread, breads/spreads; SweetBread, bread products; ChzPizzaBurger, cheese and pizza/hamburger; GrainPowder, grain powder; SnackConfect, snack/confections; NutsRoastBeans, nuts and roasted beans; SoybeanSoup, soybean paste soup; TofuSprouts, tofu/bean sprouts; Eggs, eggs; Starch, starch; Potatoes, potatoes; Baechukimchi, baechukimchi; OtherKimchi, other kimchi; SaltFermented, salt-fermented food; CabbageRadishSoup, cabbage/radish soup; VegDish, vegetable dish; LettucePerilla, lettuce/perilla leaf; Mushrooms, mushrooms; OtherVegetables, other vegetables; SweetPumpkin, sweet pumpkin; Poultry, poultry; OrganMeatByproduct, by-products (organ meat, seonji, sundae); Cuttlefish, cuttlefish; Pork, pork; ProcessedMeatSeafood, processed meat/seafood; Beef, beef; RawFishEel, sliced raw fish and eel (special fish); DogMeat, dog meat; Fish, fish; Anchovy, anchovy; Seaweeds, seaweeds; ShellfishSeafood, shellfish seafood; MilkYogurt, milk/yogurt; CarbDrinkIcecream, carbonated drink/ice cream; SoyMilk, soy milk; Coffee, coffee; TradBeverages, traditional beverages; GreenTea, green tea; NatFruit, native fruit; NonNatFruit, non-native fruit. CHO, carbohydrate; PRO, protein; Pro_veg, vegetable protein; Pro_ani, animal protein; VitA, vitamin A; VitD, vitamin D; VitK, vitamin K; VitB1, thiamine; VitB2, riboflavin; VitB6, vitamin B6; VitB12, vitamin B12; VitB5, pantothenic acid; VitB7, biotin; Ca, calcium; Ca_veg, vegetable calcium; Ca_ani, animal calcium; P, phosphorus; Na, sodium; Cl, chlorine; K, potassium; Mg, magnesium; Fe, iron; Fe_veg, vegetable iron; Fe_ani, animal iron; Zn, zinc; Cu, copper; F, fluorine; Mn, manganese; Se, selenium; Chol, cholesterol; cFiber, cereal fiber; vegeFiber, vegetable fiber; GL, glycemic load; MUFA, monounsaturated fatty acid; PUFA, polyunsaturated fatty acid; *n*3_PUFA, *n*-3 polyunsaturated fatty acid; *n*6_PUFA, *n*-6 polyunsaturated fatty acid; SFA, saturated fatty acid; Pro_soy, soy protein; GI, glycemic index; AA_BCAAs, isoleucine, leucine, valine; VitC, vitamin C; VitE, vitamin E; PA, proanthocyanidins; FLA, total flavonoids. Food items are represented by their abbreviated names (e.g., WhiteRice for cooked white rice, RiceBeans for cooked rice with beans). A total of 109 dietary factors were analyzed in this study, and consistent abbreviations are used for items that appear multiple times across different figures or tables.

**Table 1 nutrients-16-03798-t001:** General characteristics of the MRCohort and the ARIRANG and Kanghwa cohorts.

	MRCohort	ARIRANG	Kanghwa
*n*	8302	4990	3374
Age, years	61.3 ± 9.8	54.6 ± 8.3	55.9 ± 8.8
Sex, female, %	63.6	37.7	37.4
Higher education, % *	20.5	35.9	34.5
Regular exercise, % *^†^	21.5	20.1	24.3
Current smokers, %	15.0	16.2	13.4
Alcohol consumption, g/day	10.1 ± 30.1	9.5 ± 23.6	8.7 ± 27.1
Body mass index, kg/m^2^	24.2 ± 3.1	24.4 ± 3.1	24.5 ± 3.0
Total Energy intake, Kcal/day	1499 ± 402	1615 ± 426	1632 ± 452

* 12 years of schooling or more. ^†^ ≥3 times/week for ≥30 min/session. Abbreviations: MRCohort: Multi-Rural Communities Cohort; ARIRANG: Atherosclerosis Risk of a Rural Area Korean General Population.

**Table 2 nutrients-16-03798-t002:** Incidence rate ratios (IRRs) and 95% confidence intervals (CIs) of 11 validated foods associated with T2D in the three cohorts.

Food Group (Abbreviation) *	MRCohort			ARIRANG			Kanghwa		
	IRR	Original *p*-Value	FDR *p*-value ^†^	IRR	Original *p*-Value	FDR *p*-Value ^†^	IRR	Original *p*-Value	FDR *p*-Value ^†^
**Cooked rice with beans**—RiceBeans	0.37 (0.31–0.46)	5.53 × 10^−22^	6.03 × 10^−20^	0.36 (0.27–0.48)	4.84 × 10^−12^	5.28 × 10^−10^	0.37 (0.25–0.54)	2.00 × 10^−7^	2.18 × 10^−5^
**Rice cakes**—RiceCakes	0.64 (0.52–0.78)	1.01 × 10^−5^	1.83 × 10^−4^				0.48 (0.31–0.74)	5.20 × 10^−6^	2.83 × 10^−4^
**Breads/spreads**—BreadSpread	0.74 (0.60–0.91)	3.85 × 10^−3^	2.62 × 10^−2^	0.57 (0.40–0.81)	1.47 × 10^−3^	2.29 × 10^−2^	0.53 (0.36–0.78)	1.02 × 10^−3^	1.86 × 10^−2^
**Bread products**—BreadProduct	0.57 (0.47–0.71)	3.00 × 10^−7^	1.09 × 10^−5^	0.58 (0.42–0.80)	9.00 × 10^−4^	1.64 × 10^−2^	0.49 (0.32–0.75)	1.29 × 10^−3^	2.01 × 10^−2^
**Cheese and pizza/hamburger**—ChzPizzaBurger	0.72 (0.58–0.89)	2.37 × 10^−3^	1.72 × 10^−2^	0.46 (0.32–0.66)	3.78 × 10^−5^	2.06 × 10^−3^	0.50 (0.33–0.75)	8.68 × 10^−4^	1.86 × 10^−2^
**Grain powder**—GrainPowder	0.71 (0.58–0.87)	1.08 × 10^−3^	9.78 × 10^−3^	0.61 (0.44–0.84)	2.76 × 10^−3^	2.50 × 10^−2^			
**Snack/confections**—SnackConfect	0.58 (0.47–0.73)	1.50 × 10^−6^	4.09 × 10^−5^	0.52 (0.38–0.71)	5.71 × 10^−5^	2.07 × 10^−3^	0.55 (0.37–0.81)	8.30 × 10^−4^	1.86 × 10^−2^
**Nuts and roasted beans**—NutsRoastBeans	0.75 (0.61–0.93)	7.15 × 10^−3^	3.90 × 10^−2^				0.49 (0.34–0.73)	2.37 × 10^−3^	2.88 × 10^−2^
**Soy milk**—SoyMilk	0.64 (0.52–0.79)	2.48 × 10^−5^	3.37 × 10^−4^	0.62 (0.45–0.84)	2.37 × 10^−3^	2.50 × 10^−2^	0.54 (0.37–0.79)	3.89 × 10^−4^	1.41 × 10^−2^
**Traditional beverages**—TradBeverages	0.56 (0.45–0.69)	5.29 × 10^−8^	2.88 × 10^−6^	0.58 (0.42–0.80)	7.38 × 10^−4^	1.61 × 10^−2^	0.39 (0.26–0.59)	1.71 × 10^−3^	2.33 × 10^−2^
**Non-native fruit**—NonNatFruit	0.65 (0.53–0.81)	1.15 × 10^−4^	1.26 × 10^−3^	0.62 (0.44–0.86)	4.21 × 10^−3^	3.53 × 10^−2^			

* Data are expressed as IRR (95% CI). The multivariable model was adjusted for age (years), higher education level (≥12 years of education), regular exercise (≥3 times/week and ≥30 min/session), current drinking status (yes or no), alcohol consumption (g/day), and total energy intake (kcal/day). ^†^ FDR-adjusted *p*-values were calculated by comparing the highest and lowest tertiles. Foods with FDR < 5.00 × 10^−2^ in at least two of the three cohorts were considered validated.

**Table 3 nutrients-16-03798-t003:** Incidence rate ratios (IRRs) and 95% confidence intervals (CIs) for type 2 diabetes (T2D) incidence associated with 11 validated foods after mutual adjustment within clusters in the three cohorts.

Multivariable Model *	Validated Foods (Serving/Day)	T1	T2	T3	Original *p*-Value for T3	Mutually Adjusted *p*-Value for T3 ^†^
**MRCohort**						
**Cluster 2**						
**[Group 7]** Fusion diet combining traditional and non-traditional foods	TradBeverages	1.00	0.81 (0.65–0.99)	0.63 (0.51–0.79)	5.29 × 10^−8^	**3.98 × 10^−5^**
	BreadProduct	1.00	0.88 (0.71–1.09)	0.72 (0.57–0.91)	3.00 × 10^−7^	**6.74 × 10^−3^**
	SnackConfect	1.00	0.88 (0.72–1.08)	0.70 (0.56–0.88)	1.50 × 10^−6^	**2.65 × 10^−3^**
	RiceCakes	1.00	0.66 (0.54–0.82)	0.79 (0.63–0.99)	1.01 × 10^−5^	3.80 × 10^−2^
	ChzPizzaBurger	1.00	0.81 (0.65–1.02)	0.97 (0.76–1.25)	2.37 × 10^−3^	8.31 × 10^−1^
	BreadSpread	1.00	0.94 (0.74–1.19)	1.01 (0.80–1.26)	3.85 × 10^−3^	9.53 × 10^−1^
**Cluster 3**						
**[Group 3]** Diverse plant-based foods and seafood	SoyMilk	1.00	0.77 (0.62–0.95)	0.74 (0.59–0.93)	2.48 × 10^−5^	8.46 × 10^−3^
	**NonNatFruit**	**1.00**	**0.80 (0.65–0.98)**	**0.73 (0.58–0.91)**	**1.15 × 10^−4^**	**5.91 × 10^−3^**
	GrainPowder	1.00	0.85 (0.68–1.05)	0.83 (0.67–1.03)	1.08 × 10^−3^	9.56 × 10^−2^
	NutsRoastBeans	1.00	0.86 (0.69–1.05)	0.85 (0.69–1.06)	7.15 × 10^−3^	1.49 × 10^−1^
**Cluster 5**						
**[Group 4]** Rice with beans and related nutrients	**RiceBeans**	**1.00**	**0.34 (0.27–0.41)**	**0.37 (0.31–0.46)**	**5.53 × 10^−22^**	**5.53 × 10^−22^**
**ARIRANG**						
**Cluster 2**						
**[Group 3b]** Diverse plant-based foods and Seafood	**NonNatFruit**	**1.00**	**0.73 (0.54–0.98)**	**0.62 (0.44–0.86)**	**4.21 × 10^−3^**	**4.21 × 10^−3^**
**Cluster 3**						
**[Group 3a]** Diverse Plant-Based Foods and Seafood	GrainPowder	1.00	0.74 (0.55–1.00)	0.61 (0.44–0.84)	2.76 × 10^−3^	**2.76 × 10^−3^**
**Cluster 5**						
**[Group 7]** Fusion diet combining traditional and non-traditional foods	**RiceBeans**	**1.00**	**0.25 (0.19–0.36)**	**0.41 (0.31–0.56)**	**4.84 × 10^−12^**	**1.51 × 10^−8^**
	BreadSpread	1.00	1.32 (0.93–1.86)	0.96 (0.64–1.44)	1.47 × 10^−3^	8.39 × 10^−1^
	BreadProduct	1.00	0.70 (0.50–0.98)	0.83 (0.57–1.21)	9.00 × 10^−4^	3.27 × 10^−1^
	ChzPizzaBurger	1.00	0.94 (0.67–1.33)	0.71 (0.48–1.06)	3.78 × 10^−5^	9.30 × 10^−2^
	SnackConfect	1.00	0.93 (0.68–1.27)	0.72 (0.51–1.02)	5.71 × 10^−5^	6.31 × 10^−2^
	SoyMilk	1.00	0.90 (0.64–1.26)	0.88 (0.63–1.24)	2.37 × 10^−3^	4.68 × 10^−1^
	TradBeverages	1.00	0.82 (0.60–1.01)	0.80 (0.58–1.10)	7.38 × 10^−4^	1.74 × 10^−1^
**Kanghwa**						
**Cluster 2**						
**[Group 7]** Fusion diet combining traditional and non-traditional foods	RiceCakes	1.00	0.60 (0.42–0.87)	0.50 (0.33–0.76)	5.20 × 10^−6^	**1.29 × 10^−3^**
	BreadSpread	1.00	0.80 (0.55–1.16)	0.68 (0.43–1.07)	1.02 × 10^−3^	9.46 × 10^−2^
	SnackConfect	1.00	0.98 (0.67–1.42)	0.71 (0.45–1.12)	8.30 × 10^−4^	1.37 × 10^−1^
	BreadProduct	1.00	0.82 (0.55–1.22)	0.82 (0.53–1.28)	1.29 × 10^−3^	3.84 × 10^−1^
	TradBeverages	1.00	0.70 (0.48–1.01)	0.65 (0.44–0.96)	1.71 × 10^−3^	2.94 × 10^−2^
**Cluster 3**						
**[Group 3]** Diverse plant-based foods and Seafood	SoyMilk	1.00	0.68 (0.46–1.02)	0.59 (0.40–0.88)	3.89 × 10^−4^	9.48 × 10^−3^
	ChzPizzaBurger	1.00	0.85 (0.57–1.26)	0.61 (0.40–0.94)	8.68 × 10^−4^	2.61 × 10^−2^
	NutsRoastBeans	1.00	0.82 (0.57–1.18)	0.66 (0.45–0.98)	2.37 × 10^−3^	4.11 × 10^−2^
**Cluster 5**						
**[Group 4]** Rice with beans and related nutrients	**RiceBeans**	**1.00**	**0.38 (0.26–0.56)**	**0.37 (0.25–0.54)**	**2.00 × 10^−7^**	**2.00 × 10^−7^**

* Data are expressed as IRR (95% CI). The multivariable model was adjusted for age (years), higher education level (≥12 years of education), regular exercise (≥3 times/week and ≥30 min/session), current drinking status (yes or no), alcohol consumption (g/day), and total energy intake (kcal/day). ^†^ The significance threshold was set to the largest original *p*-value (7.15 × 10^−3^) for the validated dietary factors to maintain consistency with the screening step.

## Data Availability

Data described in the manuscript, code book, and analytic code will not be made available because of the data protection regulations of the Korea Disease Control and Prevention Agency (KCDA) and the cohorts involved.

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
