# Peer review of "Diet-Wide Association Study for the Incidence of Type 2 Diabetes in Three Population-Based Cohorts"

_nutrients, 2024, doi:10.3390/nu16223798_

Round 1
Reviewer 1 Report
Comments and Suggestions for Authors
1. Some text fragments are vaguely formulated, and the reader cannot seize the idea underneath them. Some examples here:
- lines 51 and 61-62: what do the “mechanistic insights crucial for understanding T2D risk” and “the complex relationships between closely related dietary components” actually refer to?
- the phrase in lines 62-64 (“While statistical adjustments like false discovery rate (FDR) in the DWAS can partially address the issue of multiple comparisons, they do not resolve the failure to account for these interdependencies.”) is imprecise; it should be developed into a larger and clearer fragment to genuinely inform the reader about its meaning.
- the significant cultural differences between the dietary patterns of Asian and Western populations and the diabetes-targeting DWAS that has already been conducted in Western countries are two completely separate issues that should not be connected within the same phrase; their reunion introduced by the “despite” preposition (lines 64-66) is illogical.
2. What regulatory institutions issued the ethics approvals for the three cohorts? The term “the respective institutions” (line 100) becomes vague when used in the context of multi-regional cohorts.
3. Lines 211-219: all abbreviations should be explained at first mention in the main text and not only in Supplementary Table 1 or below Figure 1.
Comments on the Quality of English Language6. A series of typing and syntactic errors impair the background of a relatively good-quality English language. I mainly refer here to verb-noun disagreements generated by the use of singular verb forms in contexts obviously referring to plurals, such as “DWAS on diabetes has been conducted in Western contexts [8,9]” (line 66) or “Previous DWAS on T2D has provided valuable insights” (line 354), but also to some typing omissions such as “seven cluster” (line 305) or “nin foods” (line 380).
Author Response
- Response to reviewer #1
Comment 1-1: Some text fragments are vaguely formulated, and the reader cannot seize the idea underneath them. Some examples here: (1) lines 51 and 61-62: what do the “mechanistic insights crucial for understanding T2D risk” and “the complex relationships between closely related dietary components” actually refer to?. |
Response 1-1:
Thank you for this important comment. These phrases needed more specific explanation. We have revised these sections to be more explicit about their meanings (lines 44-55): ‘The critical role of diet in T2D risk is well-established [5]. Traditional research approaches have often focused on isolated dietary components. However, this reductionist method fails to capture the intricate interplay between dietary factors, and leads to fragmented and inconsistent findings across studies [6]. To address these limitations, researchers have suggested dietary pattern approaches, such as the Mediterranean diet and the Alternative Healthy Eating Index (AHEI) [7]. However, these methods aggregate foods into broad categories, potentially masking the specific impacts of individual foods and nutrients. Consequently, there is a significant gap in understanding the impacts of individual dietary components and their interactions on T2D risk in real-world eating patterns. Given the complexity of diets and the potential for synergistic and antagonistic effects between foods/nutrients, a more nuanced analytical approach is required to clarify the true association between diet and T2D development.’
Comment 1-2: |
|
(2) The phrase in lines 62-64 (“While statistical adjustments like false discovery rate (FDR) in the DWAS can partially address the issue of multiple comparisons, they do not resolve the failure to account for these interdependencies.”) is imprecise; it should be developed into a larger and clearer fragment to genuinely inform the reader about its meaning. |
Response 1-2:
Thank you for this comment. The phrase needs to better explain the limitations of FDR adjustment in capturing dietary interactions. The revised sentence is as follows (lines 57-64): ‘DWAS allows simultaneous evaluation of numerous foods and nutrients. However, previous DWAS relied mainly on Spearman correlations between dietary factors, and presented results using Manhattan plots based on food groups. However, these analyses did not account for potential interactions between dietary factors, such as the synergistic or antagonistic effects that may occur when nutrients are consumed together. [8,9]. While statistical adjustments like false discovery rate (FDR) in the DWAS can partially address the issue of multiple comparisons, they do not resolve the failure to account for these interrelationships.’
Comment 1-3: (3) The significant cultural differences between the dietary patterns of Asian and Western populations and the diabetes-targeting DWAS that has already been conducted in Western countries are two completely separate issues that should not be connected within the same phrase; their reunion introduced by the “despite” preposition (lines 64-66) is illogical. |
Response 1-3:
I agree that these are distinct points that should be presented separately. The revised sentence is as follows (lines 64-68): ‘Currently, DWAS for diabetes have been conducted only in Western populations [8,9]. There is a need for similar studies in Asian populations because differences in food cultures and health profiles between Western and Asian populations may result in different dietary impacts on diabetes risk.’
Comment 2: |
|
What regulatory institutions issued the ethics approvals for the three cohorts? The term “the respective institutions” (line 100) becomes vague when used in the context of multi-regional cohorts. |
Response 2:
We agree that specifying the institutions provides greater transparency. The revised sentence is as follows (lines 98-103): ‘The study was conducted following the principles of the Declaration of Helsinki, and the study protocol was approved by the following Institutional Review Boards: Hanyang University, Chonnam University, and Keimyung University for the MRCohort; Yonsei Wonju College of Medicine for the ARIRANG; and Yonsei University for the Kanghwa cohort. All participants provided written informed consent before participating in the study’
Comment 3: |
|
Lines 211-219: all abbreviations should be explained at first mention in the main text and not only in Supplementary Table 1 or below Figure 1. |
Response 3:
We agree that abbreviations should be defined at their first mention in the main text and have revised the manuscript accordingly, ensuring that all abbreviations are defined upon first use in the main text.
Comment 6: |
|
6. A series of typing and syntactic errors impair the background of a relatively good-quality English language. I mainly refer here to verb-noun disagreements generated by the use of singular verb forms in contexts obviously referring to plurals, such as “DWAS on diabetes has been conducted in Western contexts [8,9]” (line 66) or “Previous DWAS on T2D has provided valuable insights” (line 354), but also to some typing omissions such as “seven cluster” (line 305) or “nin foods” (line 380). |
Response 6:
Thank you for identifying these errors. Line 66 has already been addressed in our response to Comment 1-3. For the remaining errors, we have made the following corrections:
- Line 354 (now Line 355): We changed ‘Previous DWAS on T2D has provided valuable insights’ to ‘Previous DWAS on T2D have provided valuable insights’ to maintain proper subject-verb agreement with the plural form of DWAS studies.
- Line 305 (now Line 307): We revised ‘seven cluster’ to 'seven clusters’ to ensure numerical agreement.
- Line 380 (now Line 382): We corrected the typographical error ‘nin foods’ to ‘nine other foods’ for accuracy.
Following these corrections, we conducted a thorough review of the entire manuscript to identify and correct any similar errors, ensuring consistent grammar and spelling throughout the document.
Reviewer 2 Report
Comments and Suggestions for Authors
Thank you for the opportunity of reviewing this very original and interesting article, which reports the results obtained by the Diet-Wide Association Study (DWAS) approach on 16,666 participants without T2D from 3 Asian cohorts, to investigate the links between various dietary factors and T2D onset, when considering complex dietary patterns. Eleven food clusters resulted inversely associated with T2D risk, and two of them (cooked rice with beans and non-native fruits) remained significantly inversely associated with T2D risk in more than one cohort, suggesting that incorporating beans into rice and eating various fruits may be an effective strategy for diabetes prevention.
Introduction: clearly exposes the background and the aim of the study
Materials and Method: thoroughly described
Results: they are strong, because seven dietary clusters emerged with consistency, despite the geographic differences among the three cohorts. The text is well written and refers to the tables and figures which are very detailed.
Discussion: it proposes valid elements for discussion and comparisons with the currently available literature and warrants for further research using this approach.
line 331 “such as plant-based protein” change in “such as plant protein”
Line 380: there’s a typo
Author Response
- Response to reviewer #2
Comment 7: line 331 “such as plant-based protein” change in “such as plant protein” Line 380: there’s a typo |
Response 7:
Thank you for identifying these errors. We have made the following corrections:
- Line 331 (now Line 333): We revised ‘plant-based protein’ to ‘plant protein’ for conciseness.
- Line 380 (now Line 382): We corrected the typographical error ‘nin foods’ to ‘nine other foods’ for accuracy.” This response provides clear context and specific corrections for each line.
We appreciate your very thoughtful and useful comments, again.